# Many-body phases from effective geometrical frustration and long-range interactions in a subwavelength lattice
D. Burba [1], G. Juzeliūnas [1], I. B. Spielman [2,3]✉ & L. Barbiero [4]✉

Geometrical frustration and long-range couplings are key contributors to create quantum phases with different properties throughout physics. We propose a scheme where both ingredients naturally emerge in a Raman induced subwavelength lattice. We first demonstrate that Raman-coupled multicomponent quantum gases can realize a highly versatile frustrated Hubbard Hamiltonian with long-range interactions. The deeply subwavelength lattice period leads to strong long-range interparticle repulsion with tunable range and decay. We numerically demonstrate that the combination of frustration and long-range couplings generates many-body phases of bosons, including a range of density-wave and superfluid phases with broken translational and time reversal symmetries, respectively. Our results thus represent a powerful approach for efficiently combining long-range interactions and frustration in quantum simulations.

In a diverse range of systems from neutron stars[1] to nuclei[2,3] and electrons[4,5], intrinsic strong long-range interactions are essential for stabilizing strongly correlated states[6]. Large scale quantum correlations are an essential element in a wide variety of phenomena[7]. These couplings between far-spaced elementary constituents can lead to interesting properties such as static entanglement[8,9] and symmetry breaking[10–12], as well as efficient spreading of highly non-local correlations in out-of-equilibrium configurations[13,14]. In condensed matter systems this situation is further enriched by geometric frustration which can compete with the electrons' native screened Coulomb interaction[15]. Under these conditions, a large ground state degeneracy can occur, topological[16–18] and spontaneously-symmetry-broken phases[19,20] can take place. Here we describe a 1D lattice for ultracold atoms with effective geometric frustration, and interactions extending over several lattice sites.

Our analysis is particularly relevant because the simultaneous presence of long-range couplings, geometrical frustration and quantum fluctuations challenge all current theoretical treatments[21,22]; it is, therefore, crucial to back theoretical predictions with accurate experiments. While very recent solid state experiments tackle configurations where long-range couplings coexist with kinetic frustration[23], the injection of geometrical frustration remains a distant goal. In this respect, promising initial results in specific geometries of tweezer-arrays of Rydberg atoms have been obtained[24,25], however, complimentary experimental realizations for itinerant systems such as neutral atoms in optical lattices[26,27] are lacking. Without frustration, the role of long-range interactions have been explored for magnetic-atom[28,29], polar molecules[30], and cavity QED[31] systems. Without long-range interactions,

geometrical frustration has also been experimentally investigated only in weakly interacting regime[32–37], while strong interactions have never been explored. Even theoretical proposals to engineer geometrically frustrated strongly correlated phases mainly concentrate on systems with contact interactions[38–44] or, very recently, nearest-neighbor repulsion[45]. This work provides a significant step forward by describing quantum gases in strongly interacting regimes where quantum fluctuations, geometrical frustration and long-range couplings strongly compete.

We consider the many-body physics of a recently realized class of subwavelength 1D optical lattices[46–55] and show that they are a suitable platform for combining geometric frustration and finite-ranged interactions. These lattices use Raman transitions to couple $N$ internal atomic states with lasers of wavelength $\lambda$ slightly detuned from the Raman resonance condition [Fig. 1a]. This setup is described by an extended Hubbard Hamiltonian (Bose or Fermi) where the lattice period is reduced from $\lambda/2$ to $\lambda/(2N)$. In contrast to existing optical lattice systems—such as magnetic atoms[56,57], weakly dressed Rydberg atoms[58], and polar molecules[59,60]—where the spatial decay of the interaction is fixed and tunneling processes occur between neighboring lattice sites, our lattice allows for interactions and tunneling with a tunable range. In particular, we show that: (1) the range and sign of tunneling processes can be controlled giving rise to effective geometric frustration [see Fig. 1b]; and (2) the interactions can be approximated by a power law whose exponent is a function of the Raman coupling strength.

We then turn to a specific implementation based on bosonic $^{87}$Rb and identify a range of strongly correlated regimes through a matrix-product-

¹Institute of Theoretical Physics and Astronomy, Department of Physics, Vilnius University, Saulėtekio 3, LT-10257 Vilnius, Lithuania. ²Joint Quantum Institute, University of Maryland, College Park, College Park, Maryland, 20742-4111, USA. ³National Institute of Standards and Technology, Gaithersburg, Maryland, 20899, USA. ⁴Institute for Condensed Matter Physics and Complex Systems, DISAT, Politecnico di Torino, I-10129 Torino, Italy. ✉e-mail: ian.spielman@nist.gov; luca.barbiero@polito.it

**Fig. 1 | Experimental concept. a** Lasers induce two-photon Raman transitions of intensity $\Omega$ that cyclically couple $N = 3$ consecutive internal states (labeled by $m$) with energy difference $\epsilon_m$. **b** Subwavelength lattice with long-range tunneling; all links emanating from the $j = 0$ site (with dressed state index $n = 0$ and unit cell $\ell = 0$) have their tunneling strength labeled. Top: synthetic dimension picture with triangular plaquettes and the potential for geometric frustration, with representative tunneling strengths labeled. Bottom: corresponding 1D lattice with explicit long-range links.

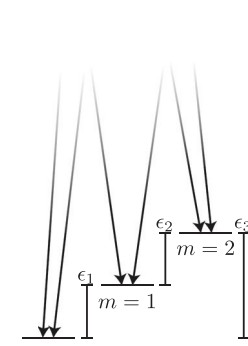

(a) Internal states

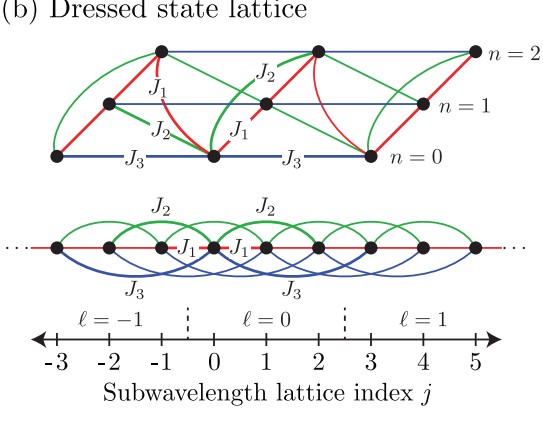

(b) Dressed state lattice

states (MPS) analysis[61]. When the $N = 3$ states of the $F = 1$ hyperfine ground state are considered, we find a normal superfluid (SF) phase in the regime of weak interaction and short range tunneling. Detuning from Raman resonance introduces geometrical frustration leading to a chiral superfluid (CSF) phase with broken time-reversal (TR) symmetry; similar CSFs have been predicted in far ranging systems from cold atoms in $p$-orbitals[62–64] to hadrons[65]. Extending the interaction range destabilizes the SF phases in favor of a spontaneous symmetry broken (SSB) density wave (DW$_{1/2}$) insulator consisting of alternating occupied and empty sites. These phases persist for $N = 5$ internal states (modeling the $F = 2$ hyperfine manifold of $^{87}$Rb). The effective lattice period decreases as $N$ increases, making long-range repulsion more significant. At filling factor 1/3 this stabilizes a period-3 density wave (DW$_{1/3}$) never achieved in cold atoms setups with individual bosons always separated by two empty lattice sites. Finally, we provide a detailed protocol for state preparation and detection, providing complete experimental access to all the interesting many-body regimes. Our results provide a solid and alternative route to explore geometrically frustrated quantum matter in presence of strong long-range correlations.

## Results
### Physical setup
We consider the one dimensional sample of ultracold bosons of mass $m_a$ illuminated by a pair of counterpropagating lasers of wavelength $\lambda$. This geometry serves to define the two photon recoil wavenumber $k_R = 2\pi/\lambda$ and energy $E_R = \hbar^2 k_R^2/(2m_a)$. The optical fields induce two photon Raman transitions that cyclically couple $N$ internal atomic states (labeled by the index $m = 0, 1, \ldots, N-1$) with strength $\Omega$. The cyclic coupling condition is fulfilled by Raman-coupling the $m = 0$ and the $m = N - 1$ states [show for $N = 3$ in Fig. 1a]. We explore the configuration of nearly-resonant Raman coupling, where each transition is detuned by a small amount $\delta_m = \epsilon_{m-1} - \hbar\delta\omega_{m-1}$ from resonance; as shown in Fig. 1a, $\epsilon_m$ is the energy difference between consecutive states $m$ and $m + 1$, and $\delta\omega_m$ is the corresponding Raman frequency-difference.

This scheme can be described by the light-matter Hamiltonian density $\hat{H}_{LM}(x) = \hat{H}_R(x) + \hat{H}_d(x)$, with contributions from Raman coupling

$$\hat{\mathcal{H}}_R(x) = -\Omega \sum_{m=0}^{N-1} e^{2ik_R x} \hat{\phi}_{m+1}^\dagger(x)\hat{\phi}_m(x) + \text{H.c.}, \tag{1}$$

and detunings

$$\hat{\mathcal{H}}_d(x) = \sum_{m=0}^{N-1} \delta_m \hat{\phi}_m^\dagger(x)\hat{\phi}_m(x), \tag{2}$$

both expressed in terms of bosonic field operators $\hat{\phi}_m^\dagger(x)$ and $\hat{\phi}_m(x)$. These describe the creation and annihilation of a particle in internal state

$m = 0, 1, \ldots, N - 1$ at position $x$. Owing to the cyclic coupling, we label the internal states periodically so that $\hat{\phi}_m^\dagger(x) = \hat{\phi}_{m+N}^\dagger(x)$, and we adopt an energy zero such that the detunings sum to zero, $\sum_m \delta_m = 0$. The operator $\hat{\mathcal{H}}_R$ effects a tight-binding lattice in a synthetic dimensional space where each internal state $m$ corresponds to a synthetic lattice site[66]. In this synthetic dimension picture, the Raman coupling in $\hat{\mathcal{H}}_R$ includes a Peierls phase $2k_R x$ on each hopping term, while the detuning term $\hat{\mathcal{H}}_d$ captures on-site energies. In analogy with conventional tight-binding lattices in real space, we rewrite $\hat{\mathcal{H}}_{LM}$ in a dressed state representation using the synthetic-dimension momentum states basis

$$\hat{\psi}_n^\dagger(x) = \frac{1}{\sqrt{N}} \sum_{m=0}^{N-1} e^{2\pi i n m/N} \hat{\phi}_m^\dagger(x), \tag{3}$$

with $n \in \{0, 1, \cdots, N-1\}$. This transformation diagonalizes the Raman coupling operator

$$\hat{\mathcal{H}}_R(x) = \sum_{n=0}^{N-1} \varepsilon_n(x)\hat{\psi}_n^\dagger(x)\hat{\psi}_n(x) \tag{4}$$

with energies

$$\varepsilon_n(x) = -2\Omega \cos\left(2k_R x - 2\pi n/N\right) \tag{5}$$

describing the usual cosinusoidal tight-binding dispersion with minima shifted from zero "crystal momentum" by the Peierls phase $2k_R x$. In terms of the real-space coordinate $x$, $\hat{H}_R(x)$ defines a set of $N$ cosinusoidal adiabatic potentials with period $\lambda/2$. The potentials corresponding to neighboring dressed states are separated from each other by a subwavelength spacing $a = \lambda/(2N)$, as illustrated in Fig. 2. The potential minima are located at spatial positions $x_j = aj$ given by the subwavelength lattice site index $j = n + N\ell$, itself defined by both the unit cell $\ell$ of the underlying $\lambda/2$ lattice as well as the dressed state $n$.

In the synthetic-dimension momentum representation the detuning Hamiltonian density

$$\hat{\mathcal{H}}_d(x) = \sum_{n,\Delta n=0}^{N-1} \gamma_{\Delta n} \hat{\psi}_{n+\Delta n}^\dagger(x)\hat{\psi}_n(x) \tag{6}$$

has off-diagonal terms that induce long-range tunneling. For odd $N$ this can be expressed in a conventional tunneling form

$$\hat{\mathcal{H}}_d(x) = \sum_{n=0}^{N-1} \sum_{\Delta n=1}^{(N-1)/2} \gamma_{\Delta n} \hat{\psi}_{n+\Delta n}^\dagger(x)\hat{\psi}_n(x) + \text{H.c.} \tag{7}$$

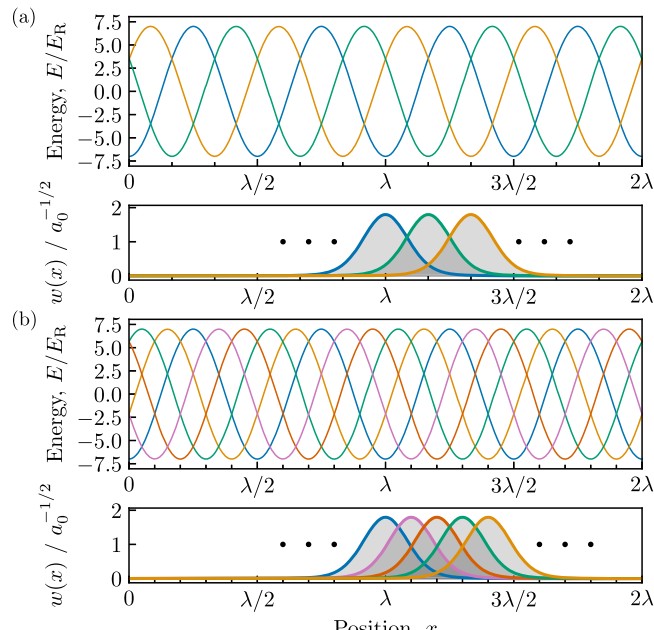

**Fig. 2 | Subwavelength Raman lattice. a, b** Lattice for $N = 3$ and $N = 5$ internal states respectively, both computed for Raman coupling $\Omega = 3.5E_R$. In both cases the top panel plots the adiabatic potentials and the bottom panel displays representative Wannier functions $w(x - ja)$; colors mark the dressed state index.

with matrix elements

$$\gamma_{\Delta n} = \frac{1}{N}\sum_{m=0}^{N-1}\delta_m e^{2\pi i m \Delta n / N} \tag{8}$$

given by a discrete Fourier transform of the detunings. For even $N$ the $\Delta n$ sum runs from 1 to $N/2$; for $\Delta n = N/2$ the tunneling matrix element must be divided by 2 to avoid double counting. In any case, these complex valued tunneling matrix elements can be expressed as the product $\gamma_{\Delta n} = |\gamma_{\Delta n}|\exp(i\phi_{\Delta n})$ of a strength $|\tilde\gamma_{\Delta n}|$ with $\gamma_0 = 0$ (since $\sum_m \delta_m = 0$) and a Peierls phase $\phi_{\Delta n}$ with $\phi_{\Delta n} = -\phi_{-\Delta n}$ (since $\delta_m$ is real valued). We focus on symmetric patterns of detuning (i.e., $\delta_m = \delta_{-m}$), in which case $\gamma_{\Delta n}$ is additionally real-valued, but still can be long-ranged with a combination of positive and negative contributions.

## Band structure and tight binding description

The preceding discussion concluded with a continuum description of our sub-wavelength lattice; to describe the many-body physics of this configuration we now construct the 1D lattice model for atoms in the lowest Bloch band. Without interactions this provides an exact description of the low-energy physics, and for large enough Raman coupling interaction-mixing of higher bands can be neglected.

In the absence of detuning, the lattice consists of $N$ independent sinusoidal potentials each with the ground-band Wannier states shown in Fig. 2. In general the operator

$$\hat{b}^\dagger_{r,n,\ell} = \int dx\, w_r^*(x - aj)\hat{\psi}_n^\dagger(x) \tag{9}$$

describes the creation of an atom in the $r$-th Bloch band of the $n$-th sublattice with Wannier amplitudes $w_r(x)$ computed for a sinusoidal-lattice[67], and as above $j = n + N\ell$. In what follows we focus on the lowest ($r = 0$) band and succinctly label these operators via $\hat{b}_j^\dagger$, using the subwavelength lattice index $j$ alone.

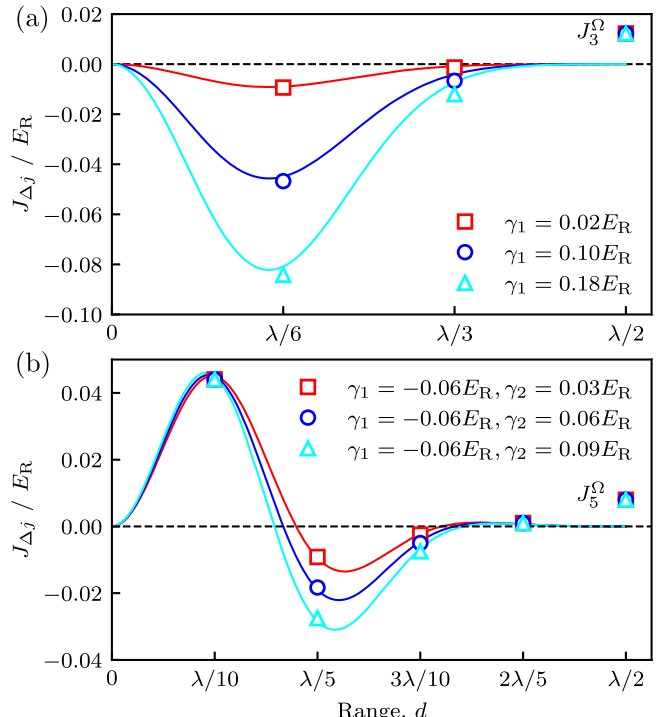

**Fig. 3 | Subwavelength tunneling $J_{\Delta j}$ as a function of distance $d$.** Computed directly from Wannier functions (markers, including native tunneling as marked), or from a Gaussian variational ansatz presented in Methods (curves, excluding native tunneling). **a** $N = 3$ internal state case, computed for Raman coupling $\Omega = 3.0E_R$ and first detuning Fourier component $\gamma_1/E_R \in \{0.02, 0.1, 0.18\}$. **b** $N = 5$ internal state case, computed for Raman coupling $\Omega = 3.5E_R$, first detuning Fourier component $\gamma_1 = -0.06E_R$ and second detuning Fourier component $\gamma_2/E_R \in \{0.03, 0.06, 0.09\}$.

The native tunneling strength within these independent lattices

$$J_N^\Omega = -\int_{-\infty}^{+\infty} dx\, w^*(x)\left[-\frac{\hbar^2\partial_x^2}{2m} + \varepsilon_0(x)\right]w(x - aN) \tag{10}$$

couples states separated by $|\Delta j| = N$ sublattice sites, depends only on the Raman coupling strength $\Omega$, and is defined to be zero for $|\Delta j| \neq N$.

Additional couplings, which are significant for distances $\Delta j < N$, are provided by detuning induced tunneling

$$J_{\Delta j}^\delta = -\gamma_{\Delta j}\int_{-\infty}^{+\infty} dx\, w^*(x)w(x - a\Delta j) \tag{11}$$

that is proportional to the overlap integral between Wannier functions associated with different atomic dressed states.

Figure 3 demonstrates that the combined tunneling $J_{\Delta j} = J_N^\Omega + J_{\Delta j}^\delta$ has a variable sign and significant long-ranged contributions; markers are computed directly from Wannier functions and curves use the Gaussian approximation and $\gamma_{\Delta j}$ (see "Methods" for details). Panel (a) illustrates the simple $N = 3$ case for three different values of $\gamma_1$ (as we note in Methods, this suffices to fully quantify the detuning induced tunneling in this case), with fixed native tunneling. This case illustrates both the long-range character of $J_{\Delta j}$ as well as the sign-inversion between short and long range. Panel (b) turns to the case of $N = 5$ internal states—specified by both $\gamma_1$ and $\gamma_2$— enabling more complicated tunneling configurations, such as shown where $\gamma_1$ and $\gamma_2$ have opposite sign.

Contrary to atoms in the ground-band of an optical lattice, where the tunneling strength—fixed by the shape of the Wannier function $w(x)$—is strictly positive and the nearest-neighbor contribution dominates, Eq. (11) along with Eq. (7) show that proper selection of detuning parameters $\delta_m$ allows for long-range tunneling with a combination of positive and negative contributions. This enables effective geometric frustration even in 1D, and in the following sections we explore the many-body interplay between effective geometric frustration and interaction processes.

### Interacting processes

The preceding section defined the single-particle tunneling contribution to our system's Hubbard model description. Here we continue by computing the two-body bosonic interactions with strength given by the overlap integral of atomic densities

$$V_{\Delta j} = g_{1D} \int_{-\infty}^{+\infty} dx\, |w(x)|^2 |w(x - a\Delta j)|^2, \quad (12)$$

where the pre-factor $g_{1D}$ describes the strength of the contact interaction that does not depend on the atomic internal state (a good approximation for ultracold $^{87}$Rb atoms in their ground electronic state[68]). Even for such simple underlying interactions, effective interactions between laser-dressed atoms generally include additional non-local contributions such as density assisted tunneling and pair tunneling (see refs. 69,70 for examples in the continuum). However, in the present case such terms are absent because the transformation in Eq. (3) is independent of the spatial coordinate and therefore leaves density-density interactions in Eq. (12) unchanged. As suggested by the Wannier orbitals in Fig. 2, the overall strength and range of $V_{\Delta j}$ can be tuned by modifying the effective lattice depth $2\Omega$ which predominantly affects the width of Wannier functions.

The detailed properties of $V_{\Delta j}$ are summarized in Fig. 4 (with numerical values suitable for $^{87}$Rb; see "Methods" for details), with markers denoting explicit numerical evaluation of Eq. (12) for $N = 3$ (red circles) and $N = 5$ (blue stars) internal states. The solid curves plot the result of a variational calculation using a Gaussian ansatz for the Wannier functions (see "Methods" for details). Panel (a) confirms that $V_{\Delta j}$ can be a significant fraction of $V_0$ over the range of a few subwavelength lattice sites, and the inset shows that, as expected for the underlying sinusoidal lattice, the overall strength depends weakly on the Raman coupling strength $\Omega$ with a nominal $V_0 \sim \Omega^{1/4}$ scaling.

### Comparison to other long-range interactions

For the usual case of ultracold atoms in the ground band of an optical lattice, the on-site interaction $V_0$ greatly exceeds longer ranged contributions because Wannier functions are highly localized to individual lattice sites. As a result, additional contributions such as dipolar interactions are required to induce long-range interactions in cold-atom systems.

In conjunction with local interactions, long-ranged interactions can be modeled by the power-law interaction potential

$$V_{\Delta j} = \delta_{\Delta j,0} V_0 + (1 - \delta_{\Delta j,0}) V_\alpha \Delta j^{-\alpha}. \quad (13)$$

In the dipolar case an applied electric or magnetic field induces interactions with $\alpha = 3$[28]; this limits the range of many-body phenomena that can be realized. Our scheme is not subject to this limitation and $\alpha$ is not fixed a priori.

For many-body physics, the very long-ranged tail of this interaction is often unimportant, making the interaction strengths at $\Delta j = 0$, 1 and 2 the only relevant contributions[71]. These can be quantified by the relative strength $V_\alpha/V_0$ of the power-law to local potentials, as well as the power law exponent $\alpha = \log_2(V_1/V_2)$. The relative strength $V_\alpha/V_0$, shown in Fig. 4(b-top), confirms that for both $N = 3$ and $N = 5$ the long-range contribution can be significant; because the interactions ultimately derive from overlap integrals, we have $1 > V_\alpha/V_0 \geq 0$. Owing to the reduced spacing between subwavelength lattice sites for increasing $N$, $V_\alpha/V_0$ is

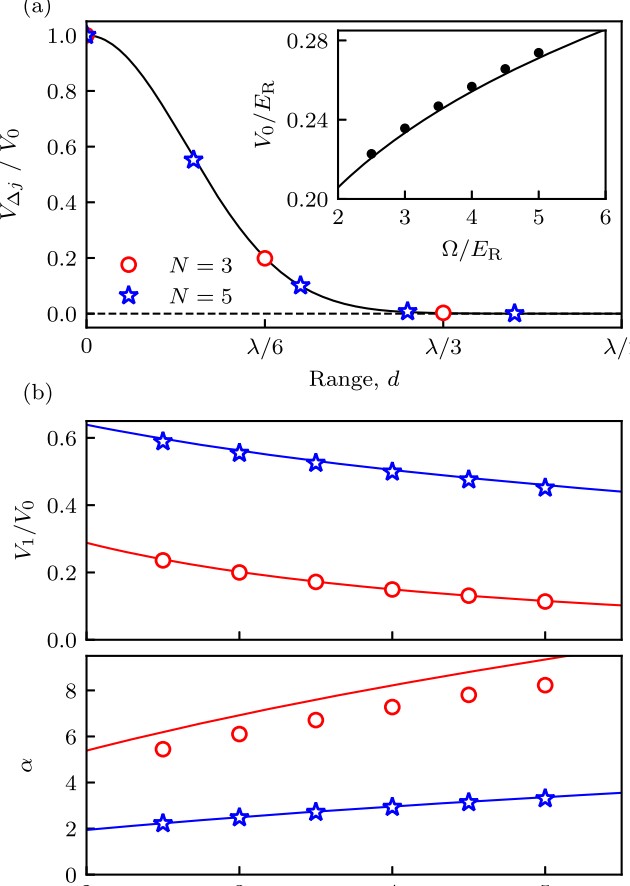

**Fig. 4 | Long-range interactions.** Computed directly from Wannier functions (markers), or from a Gaussian variational ansatz (curves). In both cases, red and blue mark $N = 3$ and $N = 5$ internal states, respectively. **a** Dependence of interaction strength $V_{\Delta j}$ on distance $d$ for Raman intensity $\Omega/E_R = 2.5$. (Inset) On-site interaction strength. **b** Top: Relative strength of long-range interactions $V_1/V_0$ as a function of Raman coupling $\Omega$. Bottom: Effective power law exponent $\alpha$ versus Raman coupling $\Omega$.

larger for $N = 5$ than $N = 3$. Figure 4(b-bottom) shows that the power law exponent is not fixed (as it would be for dipolar or Van der Waals systems), but crucially it can be tuned simply by varying $\Omega$; for our parameters $\alpha$ approximately resides in $5 \lesssim \alpha \lesssim 8$ for $N = 3$ and $2 \lesssim \alpha \lesssim 3$ for $N = 5$. This highlights the flexibility of our setup compared to conventional optical lattice realizations, and provides an avenue for realizing interesting many-body phases.

More broadly speaking, tunable power-law like scaling of the interaction's range in two-level systems has been realized for spin-spin couplings in trapped ion systems[72] and predicted for transversely confined hard-core dipolar bosons[71]. In contrast, our construction is applicable to itinerant gases of both bosonic and fermionic ultracold atoms, and, as we focus on below, it combines effective geometrical frustration in the single-particle degrees of freedom with power-law like scaling of the interactions.

Table 1 summarizes the interaction strengths $V_{0,1,2}$ for the range of Raman coupling $\Omega$ that we focus on. As compared to the typical $\lesssim k_B \times 5$ nK $= h \times 100$ Hz thermal energy scales for ultracold atoms in optical lattices, this shows that for $N = 3$ interactions are relevant for $\Delta j = 0$ and 1; for $N = 5$ the $\Delta j = 2$ interaction is also appreciable. The $\Delta j = 1$ nearest neighbor interaction strengths largely exceed those of magnetic lanthanide atoms[56,57], and instead are comparable those predicted for polar molecules[59].

**Table 1 | Interaction strength values**

| | N = 3 | | | | | | N = 5 | | | | | |
|---|---|---|---|---|---|---|---|---|---|---|---|---|
| | $V_0$ | | $V_1$ | | $V_2$ | | $V_0$ | | $V_1$ | | $V_2$ | |
| $\Omega/E_R$ | $E_R$ | $h \times$ Hz | $E_R$ | $h \times$ Hz | $E_R$ | $h \times$ Hz | $E_R$ | $h \times$ Hz | $E_R$ | $h \times$ Hz | $E_R$ | $h \times$ Hz |
| 3.0 | 0.236 | 867 | 0.047 | 172 | 0.001 | 3 | 0.236 | 867 | 0.130 | 479 | 0.024 | 87 |
| 3.5 | 0.247 | 908 | 0.042 | 155 | 0.000 | 2 | 0.247 | 908 | 0.129 | 474 | 0.020 | 74 |
| 4.0 | 0.257 | 944 | 0.038 | 140 | 0.000 | 1 | 0.257 | 944 | 0.127 | 469 | 0.017 | 63 |

Exact form of interaction strengths is given by Eq. (12). Approximate analytical expression for interaction strengths is given by Eq. (41).

## Many-body phase diagram

The combination of terms derived in the previous section can be assembled into an extended 1D Bose-Hubbard (BH) Hamiltonian

$$\hat{H} = -\sum_j \sum_{\Delta j \geq 0} J_{\Delta j}(\hat{b}_j^\dagger \hat{b}_{j+\Delta j} + \hat{b}_{j+\Delta j}^\dagger \hat{b}_j)$$
$$+ \sum_j \left[ \frac{V_0}{2}\hat{n}_j(\hat{n}_j - 1) + \sum_{\Delta j > 0} V_{\Delta j}\hat{n}_j\hat{n}_{j+\Delta j} \right], \quad (14)$$

where $\hat{n}_j = \hat{b}_j^\dagger b_j$, off resonant couplings to higher bands can be neglected in the regime of large Raman coupling ($\Omega \gtrsim 3$ for $^{87}$Rb, see "Methods" for details). For smaller Raman couplings, one would need to calculate renormalized Hamiltonian matrix elements, which arise due to higher bands[73]. While extended BH models including either geometric frustration or long ranged interactions have been widely studied[74–83], our realization embodied by Eq. (14) is the first proposal to include both long-range interactions and effective geometrical frustration. In what follows, we use MPS calculations[61] to obtain the resulting ground state phases for $N = 3$ and $N = 5$, and then we quantify the resulting phases using three quantities.

First, the staggered density

$$\delta N = \frac{1}{L}\sum_j (-1)^j(\langle \hat{n}_j \rangle - \bar{n}), \quad (15)$$

where $\bar{n} = L^{-1}\sum_j \langle \hat{n}_j \rangle$, signals period-2 density modulations, and serves as an indicator of spontaneously broken translational symmetry. In the thermodynamic limit a true period-2 SSB ground state would generally be an equally weighted superposition of these two symmetry broken configurations, making $\delta N_j = 0$; in this case a higher order correlation function would be required to extract this order. When studying this order parameter we use an odd number of lattice sites which serves to explicitly break the degeneracy between the two configurations. This order parameter would be non-zero for either density-wave solids.

Second, the single particle Green function

$$g_j^{(1)}(\Delta j) = \langle \hat{b}_{j+\Delta j}^\dagger \hat{b}_j \rangle \quad (16)$$

quantifies the degree of spatial phase coherence; in 1D, an algebraic decay of this quantity at long-range (i.e., quasi long-range off-diagonal order) reveals the presence of gapless phases with SF properties.

Lastly we consider the correlation function

$$\kappa_j^2(\Delta j) = \langle \hat{\kappa}_{j+\Delta j}\hat{\kappa}_j \rangle, \quad (17)$$

where $\hat{\kappa}_j = i(\hat{b}_j\hat{b}_{j+1}^\dagger - \hat{b}_j^\dagger\hat{b}_{j+1})$ is the local current operator for the link between $j$ and $j+1$. The long-range order of $\kappa_j^2(\Delta j)$ indicates correlations between currents on links a distance $\Delta j$ apart, and is associated with spontaneous breaking of TR symmetry. The same conclusions can be derived by calculating directly the order parameter $\hat{\kappa}_j$. Importantly, this strategy requires the addition of a weak term $\hat{\kappa}_j$ in Eq. (14) which allows breaking the ground state degeneracy associated to the currents going from left to right and vice versa with the same amplitude.

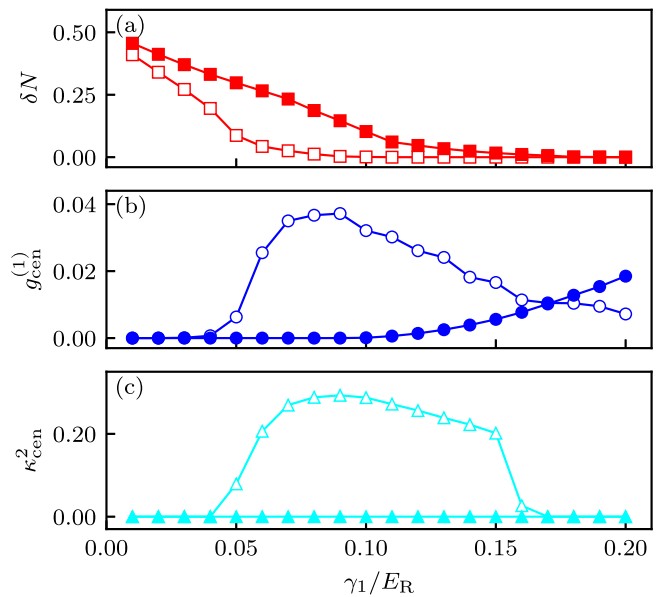

**Fig. 5 | Asymptotic correlation functions for $N = 3$ internal states as a function of detuning Fourier component $\gamma_1$. a** Staggered density $\delta N^{(1)}$; **b** One-body Green's function $g_{cen}^{(1)}$; and **c** vector order parameter $\kappa_{cen}^2$. $g_{cen}^{(1)}$ and $\kappa_{cen}^2$ are defined in Eq. (18). All three cases include Raman couplings $\Omega/E_R = 3.0$ (empty markers) and $\Omega/E_R = 3.5$ (filled markers). These were obtained in a $L = 181$ lattice site chain with 91 particles.

Our MPS calculations were performed on large systems with $L \approx 180$ sites (figure captions provide exact details), and all reported quantities were evaluated in the $L_{cen} = 100$ central sites to minimize boundary effects (from open boundary conditions). In all cases truncation errors $< 10^{-7}$ were achieved by using bond-dimensions up to 1000. We check for quasi-long range order (LRO) by evaluating correlation functions at this maximum possible range with $j_{min} = (L - L_{cen})/2$ and $\Delta j = L_{cen}$, for example, one would quantify long range phase coherence and current correlations in terms of

$$g_{cen}^{(1)} \equiv g_{j_{min}}^{(1)}(L_{cen}), \quad \text{and} \quad \kappa_{cen}^2 \equiv \kappa_{j_{min}}^2(L_{cen}). \quad (18)$$

## N = 3 internal states

We begin our MPS analysis with $N = 3$ internal states and at a fixed particle density of $\bar{n} = 1/2$ atoms per subwavelength lattice site. Geometric frustration is induced by selecting the Fourier transformed detuning $\gamma_1$ of Eq. (8) to be positive, which makes both $J_1^\delta, J_2^\delta < 0$ while the bare tunneling $J_3^\Omega$ remains, as always, positive [see Fig. 3a]. In this case, the extended BH in Eq. (14) models a triangular ladder with both ferromagnetic and anti-ferromagnetic tunnel couplings [see Fig. 1b, top panel] and long-range interactions. Figure 5a shows that staggered density order (with $\delta N > 0$) is present for small $\gamma_1$ and range of coupling strengths $\Omega$ (different curves). This indicates the presence of a SSB phase, but does not yet distinguish between supersolid and density wave (DW) insulating phases. Next, Fig. 5b shows that, by quantifying off-diagonal order, the one-body Green's

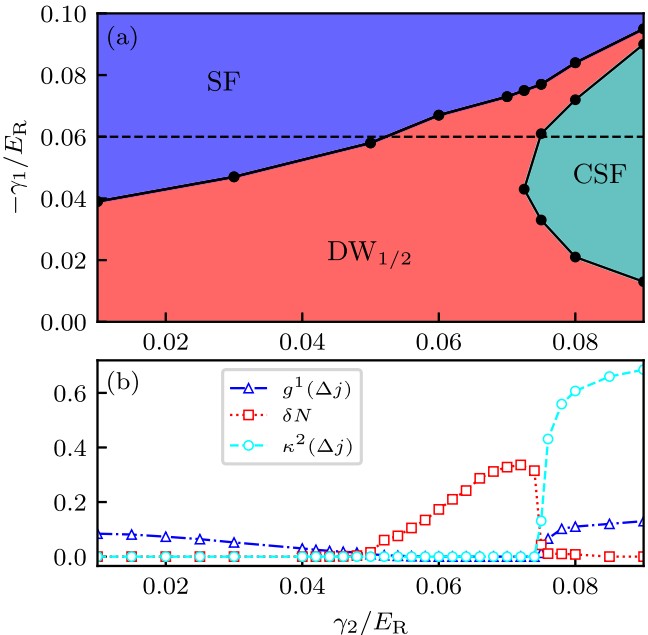

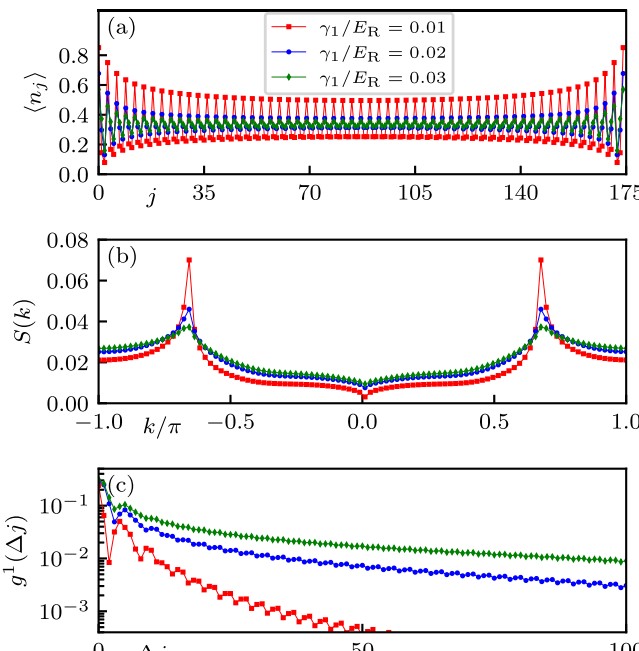

**Fig. 6 | Extended Bose-Hubbard model phase diagram and extracted values of Eqs. (15)-(18).** Both subplots were computed for $N = 5$ internal states at $\bar{n} = 1/2$ filling, Raman coupling $\Omega/E_R = 3.5$, a system size of $L = 181$ sites, and 91 particles. **a** Phase diagram as a function of detuning Fourier components $\gamma_1$ and $\gamma_2$. **b** Asymptotic correlation functions' (one-body Green's function $g_{\text{cen}}^{(1)}$, staggered density $\delta N$ and asymptotic vector order parameter $\kappa_{\text{cen}}^2$) dependence on $\gamma_2$ for fixed $\gamma_1/E_R = -0.06$, corresponding to the dashed black line in (**a**).

**Fig. 7 | Period-3 density wave order.** Computed for $N = 5$ internal states at $\bar{n} = 1/3$ filling, Raman coupling $\Omega/E_R = 3.5$, detuning Fourier component $\gamma_2/E_R = 0.01$, a system size of $L = 175$, and 59 particles. **a** Local density $\langle \hat{n}_j \rangle$; **b** structure factor $S(k)$; and **c** asymptotic single particle Green function $g_{\text{cen}}^{(1)}$.

function $g_{\text{cen}}^{(1)}$ delineates these cases. At small $\gamma_1$ there is no phase coherence, implying that the system is a DW$_{1/2}$ insulator. Changes to $\gamma_1$ proportionally change the detuning induced tunneling amplitudes (here $J_1$ and $J_2$), but have no impact on the native tunneling (given by $J_3$) nor the interaction strengths $V_{\Delta j}$. Therefore increasing $\gamma_1$ increases the kinetic contribution to the Hamiltonian, ultimately melting the DW insulator. Comparing (a) and (b) shows that $g_{\text{cen}}^{(1)}$ becomes non-negligible concurrently with the vanishing of $\delta N$; as a result we conclude that no supersolid is present and the transition is from DW$_{1/2}$ to conventional SF.

Lastly, Fig. 5c shows that the current-current correlation function $\kappa_{\text{cen}}^2$ becomes non-zero for a range of $\gamma_1$ and smaller $\Omega$ (empty markers). Comparing (b) and (c) shows that while this order parameter is only non-zero when $g_{\text{cen}}^{(1)} > 0$, the reverse is not true. This allows us to disambiguate a conventional SF phase [$g_{\text{cen}}^{(1)} > 0$ and $\kappa_{\text{cen}}^2 = 0$] from a TR broken CSF [$g_{\text{cen}}^{(1)} > 0$ and $\kappa_{\text{cen}}^2 > 0$].

In the next section, we show that by increasing the number of possible internal states from $N = 3$ to $N = 5$, CSF and more intriguing SSB phases can be engineered.

## $N = 5$ internal states

Increasing to $N = 5$ internal states offers more control over the extended BH owing to the independent tunneling parameters $\gamma_1$ and $\gamma_2$ (see "Methods" for details). Consequently, more complex configurations of tunneling amplitudes are possible. Here, we consider $\gamma_1 < 0$ and $\gamma_2 > 0$ so that $J_1^\delta, J_4^\delta > 0$ and $J_2^\delta, J_3^\delta < 0$ [see Fig. 3b]; this directly yields a geometrically frustrated lattice structure. In what follows we fix $\Omega/E_R = 3.5$, however, we verified that for $3.0 < \Omega/E_R < 4.5$ the simulation results change only quantitatively.

We first connect to our $N = 3$ results by obtaining the phase diagram in the $\gamma_1$-$\gamma_2$ plane at half filling ($\bar{n} = 1/2$) shown in Fig. 6a, and representative values of the correlation functions are shown in (b) evaluated at $\gamma_2 = -0.06$. Individual phases are identified using the logic employed in the preceding section.

For small $\gamma_1$ and $\gamma_2$ all tunneling coefficients $J_{\Delta j}$ are small compared to the long-range repulsion $V_{\Delta j}$, at half filling this stabilizes a DW$_{1/2}$ phase.

Making $\gamma_1$ increasingly negative has the predominant effect of introducing a proportionally negative $J_1$. As with the $N = 3$ case, this simply reduces the ratio $V_{\Delta j}/J_1^\delta$ of interaction to kinetic energy and melts the DW$_{1/2}$ insulator. The resulting conventional SF phase restores the bulk translational symmetry and has quasi-LRO only in $g_{\text{cen}}^{(1)}$ [see Fig. 6b]. In contrast, increasing $\gamma_2$ introduces significant effective geometrical frustration [see Fig. 3b]. The associated increased kinetic energy still destabilizes the DW$_{1/2}$ insulator, but favors a CSF where both $g_{\text{cen}}^{(1)}$ and $\kappa_{\text{cen}}^2$ are non-zero. As a result this lattice provides a unique opportunity for controlled studies of CSFs.

## Period-3 order at 1/3 filling

As is visible in Fig. 4b, the $N = 5$ long-range interactions are significant even beyond the $\Delta j = 1$ nearest neighbor scale. Repulsive interactions that are significant up to $V_{\Delta j}$ tend to favor ordered phases with a period of $\Delta j + 1$. We search for the impact of $\Delta j = 2$ next nearest neighbor interactions by reducing the particle density to $\bar{n} = 1/3$, where these interactions would favor a DW$_{1/3}$ insulator in the limit of zero tunneling. This expectation is confirmed in Fig. 7a where, at small $\gamma_2$ and for three values of $\gamma_1$, the expectation value of the on-site number operator has a period-3 oscillatory contribution.

Rather than quantify this structure in terms of a specific correlation function suited only to period-3 density order, we turn to the density-density correlation function $C_j(\Delta j) = \langle \hat{n}_{j+\Delta j} \hat{n}_j \rangle - \langle \hat{n}_{j+\Delta j} \rangle \langle \hat{n}_j \rangle$ that is sensitive to density fluctuations at a range of $\Delta j$ and its Fourier transform

$$S(k) = \frac{1}{L} \sum_{\Delta j} e^{ik\Delta j} C_j(\Delta j), \quad (19)$$

the static structure factor. Figure 7b shows that in this parameter regime the structure factor has peaks at $k = \pm 2\pi/3$ indicative of local order associated with spontaneously broken translational symmetry. The sharp peaks present for $\gamma_1 = -0.01$ are indicative of quasi-LRO, while the broad Lorentzian-like peaks for more negative $\gamma_1$ suggest an exponential decay of density-density correlations and a lack of SSB. Finally, Fig. 7(c) shows that in the small-negative $\gamma_1$ SSB case $g_j^{(1)}(\Delta j)$ vanishes exponentially, thereby confirming the presence of a DW$_{1/3}$ phase. For more negative $\gamma_1$ long-

range off-diagonal order is established suggesting a normal SF. Thus this transition (as a function of $\gamma_1$ and for small $\gamma_2$) from DW$_{1/3}$ to SF at $\bar{n} = 1/3$ filling is analogous to the DW$_{1/2}$ to SF transition at $\bar{n} = 1/2$ filling in Fig. 6a. This analysis rigorously proves that the strong long-range repulsion present in our model enables the realization of period three DW insulators, resembling $\mathbb{Z}_3$ Mott insulators predicted in chiral clock models[84].

## State preparation and detection

The previous sections identified a wide array of quantum states of matter that our setup can access. Although these phases are described by a spinless single band extended BH model, the constituent atoms exist in a dressed state representation, therefore conventional detection and measurement techniques are not effective here. In the following sections we therefore present alternative approaches to detect and prepare low energy states of our model.

## Measurement opportunities

This work has focused on distinguishing between conventional SFs, CSFs, and DW solids (where the unit-filled Mott insulator would be DW$_1$) using a range of correlation functions: the spatial density $\langle \hat{n}_i \rangle$, the static structure factor $S(k)$, the single particle Green's function $g_j^{(1)}(\Delta j)$, and the vector order parameter $\kappa_j^2(\Delta j)$. Given the deeply subwavelength nature of these lattices—$\lambda/6$ for $N = 3$ and $\lambda/10$ for $N = 5$—even today's highest resolution quantum gas microscopes[85–87] are unable to directly resolve DW order in $\langle \hat{n}_i \rangle$. Fortunately, the underlying physical structure of our system offers a unique opportunity to experimentally access our target observables.

## Standard time-of-flight

In this section, we relate momentum distributions observed in time-of-flight (ToF) images with the crystal momentum distribution $n(k)$ characterizing states in the subwavelength lattice. The crystal momentum distribution is directly related to the Fourier transformed total one-body Green's function $g^{(1)}(\Delta j) = \sum_j g_j^{(1)}(\Delta j)$ via

$$n(k) = \langle \hat{b}^\dagger(k)\hat{b}(k) \rangle = \sum_{\Delta j} e^{ik\Delta j} g^{(1)}(\Delta j), \quad (20)$$

where $\hat{b}_r^\dagger(k)$ is the creation operator for a particle with crystal momentum $k$ occupying the $r$-th band of the subwavelength lattice. Notice that $k$ is dimensionless and the Brillouin zone (BZ) extends over a range of $2\pi$.

By employing the Stern-Gerlach effect during ToF the final observed quantities are the internal state resolved momentum distributions

$$\rho_m(q) = \langle \hat{\phi}_m^\dagger(q)\hat{\phi}_m(q) \rangle,$$

where $\hat{\phi}_m^\dagger(q)$ describes the creation of a boson with wavevector $q$ in internal state $m$. In this expression $q$ has dimensions of inverse length and the subwavelength lattice BZ has an extent of $2Nk_R$; therefore we introduce a factor $c = (2\pi)/(2Nk_R)$ to convert from these physical units to the dimensionless units of the discrete lattice.

As we show in Methods, the state resolved momentum distributions are

$$\rho_m(q) = \frac{|\tilde{w}(q)|^2}{N} n[c(q - 2k_R m)], \quad (21)$$

where $\tilde{w}(q)$ is the Fourier transformed Wannier function. As such, the crystal momentum distribution, and therefore the single body Green's function $g^{(1)}(\Delta j)$, can be obtained from the internal state resolved momentum distributions, but not from the total momentum density $\rho(q) = \sum_m \rho_m(q)$.

Accessing $n(k)$ provides a powerful tool for distinguishing the many-body phases described in the previous section. Figure 8 shows that, owing to quasi LRO in $g^{(1)}(\Delta j)$, SF phases (light and dark blue) give rise to sharp peaks that vanish in insulating DW phases (red) where $g^{(1)}(\Delta j)$ decays exponentially. More specifically the normal SF exhibits a single sharp peak at $k = 0$,

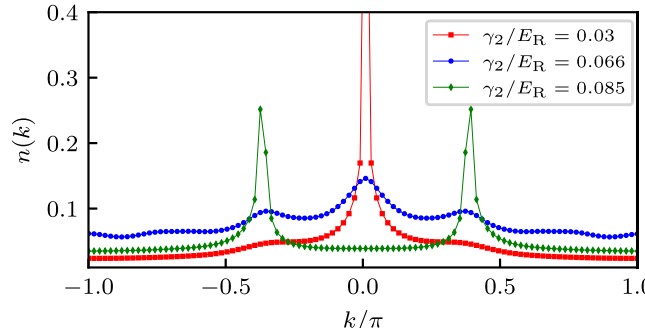

**Fig. 8 | Crystal momentum distributions $n(k)$.** Computed for Raman coupling $\Omega/E_R = 3.5$, a system size of $L = 181$, 91 particles and fixed first detuning Fourier component $\gamma_1/E_R = -0.06$ in superfluid (second detuning Fourier component $\gamma_2/E_R = 0.03$), period-2 density wave ($\gamma_2/E_R = 0.066$) and chiral superfluid ($\gamma_2/E_R = 0.085$) phases.

while the CSF has two peaks. The two peaks at incommensurate $k$ in the momentum distribution signal two minima in the dispersion relation. The interaction favors the predominant population of one of the minima and, as a consequence, the system enters a CSF phase with a non-zero local boson current characterized by a finite chirality $\langle \hat{\kappa}_j \rangle$.

These crystal momentum distributions provide little information regarding the structure of DW solids, however, higher order correlation functions do. For example, the second order function

$$n^{(2)}(\Delta k) = \int \frac{dk}{2\pi}[\langle \hat{n}(k + \Delta k)\hat{n}(k) \rangle - \langle \hat{n}(k + \Delta k)\rangle \langle \hat{n}(k) \rangle] \quad (22)$$

provides direct information regarding density order in gapped solids[88–90].

## Staggered readout

An alternate measurement protocol that is unique to this specific type of subwavelength lattice transforms each dressed state into a specific internal atomic state just prior to ToF; as above, this yields $N$ independent momentum distributions, each of which samples every $N$-th site of the subwavelength lattice (recall that $j = n + \ell N$, where $n$ is the dressed state index and $\ell$ is the $\lambda/2$ unit cell).

In order to implement this mapping we introduce two new degrees of freedom: (1) a new coupling $\Omega_{rf}$ nearly identical with the Raman coupling in Eq. (1) except that it lacks any spatial dependence [this might be implemented with a radio frequency (rf) magnetic field, or with Raman transitions in a co-propagating geometry]; and (2) a detuning proportional to the internal state index, i.e., $\delta_m = \Delta_F m$, as would be given by the usual linear Zeeman effect.

Our protocol adiabatically transforms dressed states into internal atomic states, where the adiabatic timescale $T$ is selected to be rapid as compared to the ground-band atomic dynamics, but slow compared to the band splitting. In the following, each step is correspondingly marked in Fig. 9a:

(i)  In the first step we quench to zero the detunings $\delta_m$ (used to generate long-range tunneling) in a timescale $\tau$, which is rapid compared to the adiabatic timescale $T$ used for the following steps. An adiabatic timescale would cause detuning-induced Rabi oscillations, which would ruin the one-to-one mapping between dressed and bare states. Since the detuning quench is fast, it is not shown in Fig. 9a. This step returns the system to $N$ interpenetrating but decoupled lattices.

(ii)  Next, the spatially uniform coupling $\Omega_{rf}$ is ramped on while the the Raman coupling $\Omega$ is simultaneously ramped off. This transforms each independent Raman lattice with energy $-2\Omega \cos(2\pi n/N - 2k_R x)$ into a spatially uniform dressed state with energy $-2\Omega_{rf} \cos(2\pi n/N)$.

(iii)  Lastly, $\Omega_{rf}$ is ramped off while the conventional detuning is ramped to a final value of $\Delta$.

We therefore conclude that this process transforms states in the $|n\rangle$ dressed state into the $|j = n\rangle$ internal atomic state.

**Fig. 9 | Temporal dependence of various quantities during staggered readout.** Mapping from dressed state to bare state is performed for times $t \in [0, 2T]$. **a** Ramping of different system parameters: (i) Raman coupling $\Omega$, (ii) Position independent coupling $\Omega_{rf}$, (iii) Detuning $\Delta_F$. **b** Dynamics of dressed state population. **c** Dynamics of bare state population.

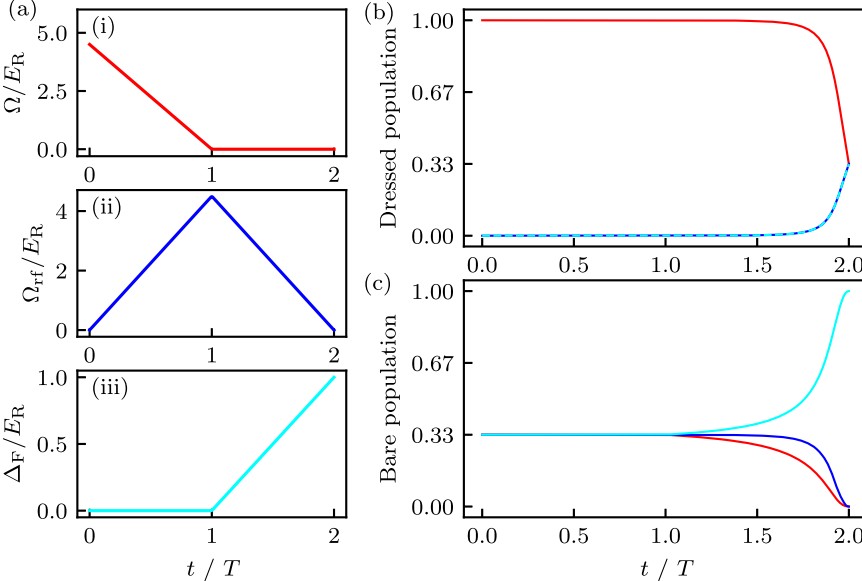

## Conclusions

In this manuscript, we showed that a recently realized class of 1D subwavelength optical lattices[46–50] lead to extended BH models with effective geometric frustration and long-ranged interactions. These lattices Raman couple internal atomic states, and a lattice potential emerges in a dressed state basis whose spacing is reduced by a factor equal to the number of coupled internal states. This configuration features significant interparticle repulsion over the scale of several lattice sites, in the absence of dipolar or Coulomb couplings. On short scales the functional form can be approximated as local repulsion combined with a power-law tail, the exponent of which can range from $-2$ to $-8$ depending on the Raman coupling strength and the number of internal states. Tunneling on the subwavelength scale is induced by small deviations from the Raman resonance condition; specifically, tuning the detuning parameters modifies the sign, Peierls phase, and strength of tunneling connecting lattice sites spaced by distances equal to number of coupled internal states.

Controlling the relative sign of the tunneling amplitude at different ranges leads to regimes of effective geometrical frustration. This is in contrast with conventional optical lattice platforms where the range and sign of hopping processes are fixed. Other approaches inducing geometric frustration by periodically modulating one or more parameter in the single particle Hamiltonian[91] with interactions, this process induces many-body dephasing which manifests as heating and then atom loss. In the present work, spontaneous emission from the Raman lasers is the only intrinsic heating process; for the example presented here, the associated $1/e$ lifetimes would be about 500 ms or $\approx 400 \times (hV_0)^{-1}$[92], enabling quantum fluctuations to remain stable for long times.

We explored the many-body potentialities offered by this setup with a detailed numerical analysis using matrix product states across a range of interactions and frustration. We found that geometrical frustration favors CSFs with spontaneously broken TR symmetry; these are of great interest both in condensed matter[62] and high energy[65] physics. The competing presence of strong long-range repulsion favored insulating DWs[1,93,94] with spontaneous spatial symmetry breaking.

Although the parameters used in matrix product states analysis are specific to $^{87}$Rb, our laser-coupling scheme can be applied to a large variety of atomic species including both bosons and fermions. In the case of fermions, the extended Hubbard model analogous to Eq. (14) still describes spinless particles; owing to Pauli repulsion, only long-range interactions with $p$-wave (and higher order) character are present[70]. Embedding such a fermionic subwavelength system in a Bose-Einstein condensate would be an intriguing next step. In analogy with materials with phonon mediated interactions between electrons, we expect the emergence of an oscillating bosonic-mediated interaction between fermions of Ruderman-Kittel-Kasuya-Yosida (RKKY)-type[95–97]. Noticeably, cold atomic systems have only been able to present preliminary results on the possible appearance of fermionic-mediated RKKY-type interactions between bosons[98,99]. Our proposed setup thus poses itself as a relevant source to engineer complex interacting processes analogous of real materials.

Figure 4 modeled interactions in our lattice as an effective power-law, valid for the first three subwavelength lattice sites. A similar procedure can be followed to instead frame these interaction terms as a screened Coulomb interaction, for example, of the repulsive Yukawa form, $V_j = V_y e^{-\gamma_y j}/j$ with $\gamma_y = \ln[V_1/(2V_2)]$. Therefore, our lattice might be applicable for quantum simulation of plasma physics[100] and screened electronic systems[101]. We considered interactions of the SU($N$) that were the same for all atomic internal states. This can be a good approximation in some cases (such as $^{87}$Rb atoms) and nearly exact in others (such as $^{87}$Sr and $^{173}$Yb)[102]. In other cases the interaction strengths can differ greatly; for example, Feshbach resonances can induce significant differences[103]. This would result in non-local interaction driven tunneling processes like pair hopping that give rise to pair superfluids (PSFs) in bosonic models[104–106].

In conclusion, our results represent an alternative proposal which can finally shed light on the investigation of long-range frustrated quantum systems.

## Methods
### Physical parameters
Here we summarize the explicit numerical values of the physical parameters used in our many-body calculations (all taken for $^{87}$Rb).

The one-dimensional interaction strength[107] is

$$g_{1D} = \frac{4\hbar^2 a_{22}}{ma_\perp^2}\left(1 - C\frac{a_{22}}{a_\perp}\right)^{-1}$$
$$= 1.05 \times 10^{-37}\,\text{J m},\tag{23}$$

having used the constant $C \approx 1.4603$[107], the reduced Planck's constant $\hbar = 1.05 \times 10^{-34}$ m$^2$kg/s, the atomic mass $m = 86.9$ AMU $= 1.44 \times 10^{-25}$ kg,

the $F = 2$ manifold $s$-wave scattering length $a_{22} = 95a_{\text{Bohr}} = 5.02 \times 10^{-9}$ m[108] and transverse confinement length $a_\perp = 1.25 \times 10^{-7}$ m. We used the aforementioned value of $g_{1D}$ for both $N = 3$ and $N = 5$. This is reasonable since $a_{11} = 100a_{\text{Bohr}}$ and $a_{22} = 95a_{\text{Bohr}}$ are fairly close to one another and thus the presented results will not qualitatively change.

The single photon recoil energy

$$E_{\text{R}} = \frac{\hbar^2 k_{\text{R}}^2}{2m} = 2.437 \times 10^{-30} \text{ J} \tag{24}$$

$$= h \times 3678 \text{ Hz}, \tag{25}$$

and wavevector $k_{\text{R}} = 2\pi/\lambda$ both require additional knowledge of the optical wavelength (here $\lambda = 790$ nm) used to create the lattice potential.

## Interaction Hamiltonian

Here we consider properties of the interaction Hamiltonian for the case of state independent [sometimes called SU($N$)] interactions. This makes the interaction Hamiltonian a function of the total local density $\hat{n}_{\text{tot}}(x) = \sum_m \hat{\phi}_m^\dagger(x)\hat{\phi}_m(x) = \sum_n \hat{\psi}_n^\dagger(x)\hat{\psi}_n(x)$, which takes the same form in the bare atomic basis (with creation operators $\psi_n^\dagger(x)$) and the dressed basis (with creation operators $\phi_m^\dagger(x)$). As a result, the interaction Hamiltonian is also unchanged with

$$\hat{H}_{\text{int}} = \frac{g}{2} \int dx : \hat{n}_{\text{tot}}^2(x) :$$
$$= \frac{g}{2} \sum_{m,m'} \int dx \, \hat{\phi}_m^\dagger(x)\hat{\phi}_{m'}^\dagger(x)\hat{\phi}_{m'}(x)\hat{\phi}_m(x) \tag{26}$$

$$= \frac{g}{2} \sum_{n,n'} \int dx \, \hat{\psi}_n^\dagger(x)\hat{\psi}_{n'}^\dagger(x)\hat{\psi}_{n'}(x)\hat{\psi}_n(x), \tag{27}$$

where $: \cdots :$ denotes the normal ordering operation.

Expanding the dressed field operators in terms of lowest band Wannier functions

$$\hat{\psi}_n^\dagger(x) = \sum_l w(x - la_0 - na)\hat{b}_{Nl+n}^\dagger \tag{28}$$

leads directly to the density-density interaction

$$\hat{H}_{\text{int}} = \sum_j \left[ \frac{V_0}{2} \hat{n}_j(\hat{n}_j - 1) + \sum_{\Delta j > 0} V_{\Delta j} \hat{n}_j \hat{n}_{j+\Delta j} \right] \tag{29}$$

that appeared in the Hubbard model [Eq. (14)], where $j = Nl + n$ denotes lattice site index and $\Delta j$ denotes distance between lattice sites.

Additional terms, such as density-induced tunneling (DIT), can appear when interaction strength $g_{j,j'}$ becomes a function $j$ and $j'$. In this case the interaction Hamiltonian

$$\hat{H}_{\text{int}}' = \frac{1}{2} \sum g_{jj'} \int dx \, \hat{\phi}_j^\dagger(x)\hat{\phi}_{j'}^\dagger(x)\hat{\phi}_{j'}(x)\hat{\phi}_j(x) \tag{30}$$

is no longer invariant with respect to a change of basis, and the dressed state Hamiltonian

$$\hat{H}_{\text{int}}' = \frac{1}{2} \sum g_{nn'mm'} \int dx \, \hat{\psi}_n^\dagger(x)\hat{\psi}_{n'}^\dagger(x)\hat{\psi}_{m'}(x)\hat{\psi}_m(x) \tag{31}$$

contains every possible combination of field operators, where

$$g_{nn'mm'} = \sum g_{jj'} \times U_{nj}^\dagger U_{n'j'}^\dagger U_{j'm'} U_{jm}. \tag{32}$$

Once again expanding the field operators in the Wannier basis, one obtains

$$\hat{H}_{\text{int}}' = \frac{1}{2} \sum_{i,j,k,l} V_{i,j,k,l} \hat{b}_i^\dagger \hat{b}_j^\dagger \hat{b}_k \hat{b}_l, \tag{33}$$

where the sum is over all subwavelength lattice sites and interaction strengths

$$V_{ijkl} = g_{i,j,k,l} \int dx \, w_i^*(x) w_j^*(x) w_k(x) w_l(x). \tag{34}$$

Coefficients such as $V_{ijki}$ lead to DIT.

In the considered experimental situation, i.e., National Institute of Standards and Technology (NIST) $^{87}$Rb cyclic coupling experiment[47], interactions are homogeneous at the 0.995 fractional level making DIT terms negligible.

## Detuning induced tunneling

As mentioned previously, the synthetic dimension tunneling parameters $\gamma_{\Delta n} = |\gamma_{\Delta n}| \exp(i\phi_{\Delta n})$ are significantly constrained by the properties of the discrete Fourier transform, as well as our restrictions on the allowed detunings:

1. $\gamma_{\Delta n} = \gamma_{\Delta n+N}$ owing to the periodicity of Fourier transforms.
2. $\gamma_0 = 0$, because $\sum_m \delta_m = 0$.
3. $\phi_{\Delta n} = -\phi_{-\Delta n}$, because the detunings $\delta_m$ are real valued.
4. For our current subwavelength lattice we focus on the simplification $\delta_m = \delta_{-m}$, making $\gamma_{\Delta n}$ real-valued and symmetric. Note that this condition is violated for our staggered readout procedure for mapping dressed states to bare states.

For example, for the $N = 3$ case, the two detuning constraints reduce the number of free degrees of freedom to one, implying that $\gamma_1$ alone quantifies the detuning induced tunneling. Similarly the $N = 5$ configuration has two independent degrees of freedom, $\gamma_1$ and $\gamma_2$.

For odd $N$ these constraints allow Eq. (8) to be simplified as

$$\gamma_{\Delta n} = \frac{1}{N} \sum_{m=0}^{N-1} \delta_m e^{2\pi i m \Delta n/N}$$
$$= \frac{2}{N} \left\{ -\left( \sum_{m=1}^{N-1} \delta_m \right) + \left[ \sum_{m=1}^{N-1} \delta_m \cos(2\pi m \Delta n/N) \right] \right\}, \tag{35}$$

where we reindexed the sum to run from $-(N-1)/2$ to $(N-1)/2$ and combined exponentials at positive and negative $m$ into cosine terms. This shows that $\delta_1, \cdots, \delta_{(N-1)/2}$ are the independent degrees of freedom. This expression can be inverted to provide a relation between a desired set of $\gamma_{\Delta n}$ and the experimental parameters $\delta_m$.

## Variational Gaussian Wannier approximation

Here we derive the approximate Wannier functions yielding the continuous curves in Fig. 4. In brief, these begin with a simple Gaussian approximation for a wavepacket centered on a single lattice site, and then we use a variational ansatz to optimize the width.

We begin with the dimensionless (with energy in units of $E_{\text{R}}$ and length in units of $k_{\text{R}}^{-1}$) Hamiltonian

$$\hat{H} = \hat{k}^2 - \frac{s}{2}[\cos(2\hat{x}) - 1] \tag{36}$$

for a particle moving in a lattice potential of depth $s = 4\Omega/E_{\text{R}}$. The second order series expansion around $x = 0$ yields the harmonic oscillator Hamiltonian

$$\hat{H} \approx \hat{k}^2 + s\hat{x}^2 \tag{37}$$

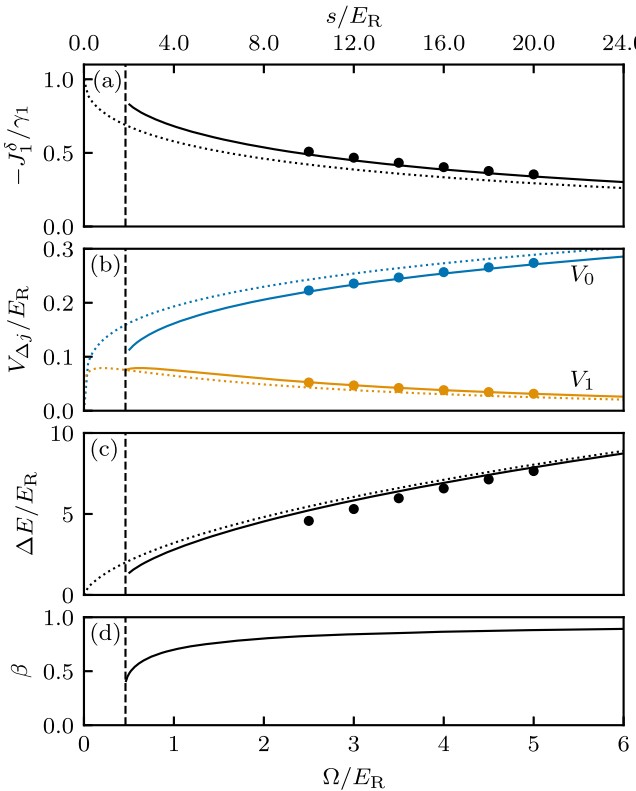

**Fig. 10 | Model parameters' dependence on Raman coupling $\Omega$.** Plotted quantities are computed directly from Wannier functions (markers), from Gaussian variational ansatz (solid curves), or from standard Gaussian ansatz (dotted curves). **a** Normalized detuning-induced nearest neighbor tunneling $-J_1^\delta/\gamma_1$. **b** Interaction strength $V_{\Delta j}$ in units of recoil energy $E_R$. **c** First energy gap $\Delta E$ in units of recoil energy $E_R$. **d** Variational parameter $\beta$. Dashed vertical lines mark the critical lattice depth $s_c = (e/2)^2$; for arguments below $s_c$ the variational parameter $\beta$ becomes complex.

with oscillator frequency $\omega = 2\sqrt{s}$, length $\ell = s^{-1/4}$, and ground state wavefunction

$$w_0(x,s) = \frac{s^{1/8}}{\pi^{1/4}} e^{-x^2\sqrt{s}/2}. \tag{38}$$

Changing Eqs. (10), (11) and (12) to dimensionless quantities, this gives explicit relations for: native tunneling

$$
\begin{aligned}
\frac{J_N^\Omega}{E_R} &= -\int dx\, w_0^*(x)\hat{H}w_0(x-\pi) \\
&= -\frac{e^{-\pi^2\sqrt{s}/4}}{4}\left[2\sqrt{s} + s\left(2 - \pi^2 + 2e^{-1/\sqrt{s}}\right)\right],
\end{aligned}
\tag{39}
$$

detuning induced tunneling

$$
\begin{aligned}
\frac{J_{\Delta j}^\delta}{\gamma_{\Delta j}} &= -\int dx\, w_0^*(x)w_0(x-d) \\
&= -e^{-d^2\sqrt{s}/4},
\end{aligned}
\tag{40}
$$

and interactions

$$
\begin{aligned}
\frac{V_{\Delta j}}{g_{1D}k_R} &= \int dx\, |w_0(x)|^2|w_0(x-d)|^2 \\
&= \frac{s^{1/4}}{\sqrt{2\pi}} e^{-d^2\sqrt{s}/2},
\end{aligned}
\tag{41}
$$

where we have introduced the dimensionless displacement $d = \pi\Delta j/N$ between the center of the Wannier orbitals. We include an expression for the native tunneling $J_N^\Omega$, but note that the Gaussian ansatz leads a nonphysical dependency on the zero of energy (because Gaussian wavepackets at neighboring lattices sites are not orthogonal).

The dashed curves in Fig. 10b plot the on-site interaction computed using this expression along with the numerically computed points. The agreement is poor.

We improved the accuracy of wavefunctions of this form using the variational principle where we replaced $s \to \beta^2 s$ and minimized the energy functional

$$
\begin{aligned}
\mathcal{E} &= \int dx\, w_0(x,\beta^2 s)\left\{-\partial_x^2 - \frac{s}{2}[\cos(2x) - 1]\right\}w_0(x,\beta^2 s) \\
&= \frac{s}{2}\left[1 + \frac{\beta}{\sqrt{s}} - \exp\left(-\frac{1}{\beta\sqrt{s}}\right)\right]
\end{aligned}
\tag{42}
$$

with respect to $\beta$. This yields the condition

$$\beta^2 = e^{-\frac{1}{\beta\sqrt{s}}}, \tag{43}$$

which is solved by

$$\beta = \exp\left[W\left(-\frac{1}{2\sqrt{s}}\right)\right], \tag{44}$$

where $W(x)$ is the notorious Lambert $W$ function[109,110]; Fig. 10d plots $\beta$ as a function of laser coupling strength $\Omega$. Because $W(x)$ becomes imaginary for arguments below $-1/e$, the expression for $\beta$ is only defined for $s > (e/2)^2$, and for large $s$, $\beta$ approaches unity confirming that the standard Gaussian approximation is accurate for very deep lattices. The solid curves in Fig. 10b show the improved on-site interaction energy computed using this correction factor.

**First excited band**

One can also accurately compute the first energy gap. To do this, we approximate the excited state Wannier function with the first excited harmonic oscillator wavefunction

$$w_1(x,s) = \sqrt{2}\frac{s^{3/8}}{\pi^{1/4}} x e^{-x^2\sqrt{s}/2}, \tag{45}$$

and in analogy with Eq. (42) we obtain the excited state energy

$$\mathcal{E}_{ex} = \frac{\sqrt{s}}{2}\left[\sqrt{s} + 3\beta + \frac{2 - \beta\sqrt{s}}{\beta}\exp\left(-\frac{1}{\beta\sqrt{s}}\right)\right].$$

Subtracting the ground state energy yields the energy gap

$$\Delta E = \frac{\sqrt{s}}{\beta}\left[\beta^2 + \exp\left(-\frac{1}{\beta\sqrt{s}}\right)\right]. \tag{46}$$

Our DMRG computations were performed for $\Omega/E_R > 3$, where the impact of higher Bloch bands of the sinusoidal adiabatic potentials are negligible. Using Eq. (46), the first band gap can be approximated as $\Delta E/E_R = 5.8$ (direct numerics give $\Delta E/E_R = 5.3 E_R$) which is large compared to the interaction scales (with $V_j/E_R \lesssim 0.25$, see Table 1), and the tunneling scales [$J_{\Delta j}/E_R \lesssim 0.5$, see Fig. 10a]. Because the many-body physics under study require temperatures $k_B T \lesssim (J_1, V_0)$, thermal excitations are also negligible.

**Relation to continuum degrees of freedom**

Here we relate the momentum distribution observed in ToF images with the crystal momentum distribution of states in the subwavelength lattice. By employing the Stern-Gerlach effect during ToF the final observed quantities

are the internal state resolved momentum density operators

$$\hat{\rho}_m(q) = \hat{\phi}_m^\dagger(q)\hat{\phi}_m(q), \tag{47}$$

where $\hat{\phi}_m^\dagger(q)$ describes the creation of a boson at momentum $q$ in internal state $m$. In terms of the continuum dressed state field operators in Eq. (3) this becomes

$$\hat{\phi}_m^\dagger(q) = \frac{1}{\sqrt{N}}\sum_n \int \mathrm{d}x\, e^{i(qx - 2\pi nm)}\hat{\psi}_n^\dagger(x), \tag{48}$$

leading to the final expression in the discrete Wannier basis

$$\hat{\phi}_m^\dagger(q) = \frac{1}{\sqrt{N}}\sum_{r,n,\ell} \int \mathrm{d}x\, e^{i(qx - 2\pi nm/N)} w_r(x_{n,l})\hat{b}_{r,n,\ell}^\dagger(x) \tag{49}$$

having made use of

$$\hat{\psi}_n^\dagger(x) = \sum_{r,\ell} w_r(x_{n,l})\hat{b}_{r,n,\ell}^\dagger, \tag{50}$$

where

$$x_{\ell,n} = x - \left(\ell + \frac{n}{N}\right)\frac{\lambda}{2}. \tag{51}$$

Here we again assume that only the ground band ($r = 0$) is relevant and thus we write

$$\hat{\phi}_m^\dagger(q) = \frac{\tilde{w}(q)}{\sqrt{N}}\sum_{n,\ell} \exp\left(i\phi_{\ell,n,m}(q)\right)\hat{b}_{n,\ell}^\dagger(x) \tag{52}$$

$$= \frac{\tilde{w}(q)}{\sqrt{N}}\sum_j \exp\left\{i\left[q - 2k_{\mathrm{R}}m\right]\frac{\lambda j}{2N}\right\}\hat{b}_j^\dagger(x) \tag{53}$$

where $\phi_{\ell,n,m}(q) = q(n + N\ell) - 2k_{\mathrm{R}}nm$ and we have made use of the periodicity of the exponential function $\exp(2\pi i nm/N) = \exp(2\pi i(n + N\ell)m/N)$. Implementing the dressed state transformation gives

$$\hat{\phi}_m^\dagger(q) = \tilde{w}(q)\hat{b}^\dagger\left(2\pi\frac{q - 2k_{\mathrm{R}}m}{2Nk_{\mathrm{R}}}\right). \tag{54}$$

Because we defined crystal momentum to be dimensionless with a BZ $2\pi$ in extent. We see that this expression links momentum $q$ in state $m$ with a crystal momentum $2\pi(2/(2Nk_{\mathrm{R}}) - m/N)$ where $2Nk_{\mathrm{R}}$ is the extent of the BZ in physical units, shifted by $2\pi m/N$ which can be interpreted as a result of the recoil kick imparted by each two-photon Raman transition.

This result connects the internal-state resolved momentum density operator and the crystal momentum density

$$\hat{\rho}_m(q) = \frac{|w(q)|^2}{N}\hat{n}(q - 2k_{\mathrm{R}}m), \tag{55}$$

and shows that the probability is split equally between the $N$ internal states.

## Data availability
Data of the study can be obtained from the corresponding authors upon request.

## Code availability
Code used in the study can be obtained from the corresponding authors upon request.

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

## Acknowledgements

We thank F. Schreck and P. Thekkeppat for discussions. L.B. acknowledges funding from Politecnico di Torino, starting package Grant No. 54 RSG21BL01 and from the Italian MUR (PRIN DiQut Grant No. 2022523NA7). I.B.S. was partially supported by the National Institute of Standards and Technology; the National Science Foundation through the Quantum Leap Challenge Institute for Robust Quantum Simulation (grant OMA-2120757); and the Air Force Office of Scientific Research Multidisciplinary University Research Initiative "RAPSYDY in Q" (FA9550-22-1-0339). D.B. and G.J. acknowledge support from the Lithuanian Research Council (Grant No. S-MIP-24-97). D.B. used resources at the High Performance Computing Center (HPCC), "HPC Sauletekis" in Vilnius University, Faculty of Physics.

## Author contributions

L.B. initiated the work. D.B. and G.J. performed theoretical analysis on the subwavelength Raman lattice. D.B. numerically calculated Wannier functions and Hamiltonian matrix elements. D.B. and G.J. explicitly derive the extended BH Hamiltonian. L.B. performed all of the quantum many-body analysis with DMRG calculations (asymptotic correlation functions, phase diagram, structure factor, etc.) on the extended BH Hamiltonian. I.B.S. considered experimental feasibility and implementation questions. I.B.S. also studied experimental state preparation and measurement details. D.B. numerically calculated dynamics during staggered readout. I.B.S. proposed variational Gaussian Wannier approximation.

## Competing interests

The authors declare no competing interests.
