## [Transparent Peer Review file · Communications Physics]

Many-body phases from effective geometrical frustration and long-range interactions in a subwavelength lattice

Corresponding Author: Mr Domantas Burba

Version 0:

Reviewer comments:

Reviewer #1

(Remarks to the Author)

Burba and coauthors studied long-range tunnelings and long-range interactions in a subwavelength lattice. They found an interesting feature that the signs of long-range tunnelings could be tuned, resulting in geometrical frustration. Since it is typically difficult to engineer geometrical frustration in ordinary optical lattices, the authors' proposal may provide experimentalists with a new useful approach to access geometrical frustration using a relatively simple setup, cyclic Raman couplings. The author further studied many-body phases including a chiral superfluid and some density waves. Whereas I think that the authors' results are interesting and sound, I believe that the authors should address a few issues before publication of this manuscript in Communications Physics.

1. It seemed that the authors did not provide an explanation for the origin of the chiral superfluid. Does it arise purely from the geometrical frustration or interactions are also necessary? The numerical result of the momentum distribution of this phase shown in Fig. 8 is very interesting since the central peak at zero momentum vanishes while two side peaks arise at finite momenta. As one could imagine that this will happen if the single-particle spectrum has two minima at finite momenta, what happens here? The authors should clarify what is the single-particle spectrum and whether interactions also play a role.

2. Chiral often means a finite current, $\langle \text{angle} \hat{k}_j \rangle$, such as the chiral edge state of a topological phase. Whereas the correlation function $k_j^2 (\Delta j)$ the authors computed is an important quantity, how about $\langle \text{angle} \hat{k}_j \rangle$ itself?

3. In ordinary optical lattices, lowering the lattice depth could also increase the amplitudes of long-range interactions due to the increase of the width of the Wannier wavefunctions. But the price to pay is the smaller band gap. As such, when long-range interactions become significant, the band gap may be small enough that p band and even higher bands cannot be ignored anymore. This is the reason that in typical optical lattices, a single-band Hubbard model includes only onsite interactions, not long-range interactions. What happens here to the band gap once the long-range interactions become large enough to induce density waves? If the interactions are comparable to the band gap, one should concern fate of results based on only the lowest band.

4. The authors missed some references about subwavelength lattices, such as PRL 122, 065303 (2019), PRL 123, 260405 (2019), PRL 126, 193001(2021) PRR 3, 023058 (2021), PRL 128, 173202 (2022).

Reviewer #2

(Remarks to the Author)

The manuscript "Many-body phases from effective geometrical frustration and long-range interactions in a subwavelength lattice" demonstrates that Raman-induced sub-wavelength lattices allow to realise Hubbard models with frustrated long-range tunnel couplings, and long-range density-density interactions. It then shows via numerical simulations that this results in a range of many-body phases, namely density wave, superfluids and chiral superfluids. The manuscript finally provides a protocol to measure observables characterising these phases, which are defined in a dressed state basis, thus, making them experimentally observable.

It therefore makes a number of significant contributions and advances. (i) it realises frustrated long-range tunable tunnel couplings/geometric frustration, typically not (easily) accessible in optical lattices, (ii) it realises long-range repulsive interactions in a system with nominally only contact interactions, (iii) it establishes a range of many-body phases, and how to

observe them experimentally.

The results are well presented and fully supported by the theoretical model and equations, the accompanying figures, and the (numerically-exact) simulation methods, and are fully convincing. The results are appropriately discussed in the context of the prior literature. Technical details are appropriately discussed in detail in the supplementary material, allowing reproduction of the results.

The work is timely and of great interest to the wider community, as sub-wavelength lattices are only beginning to be explored, and as this work shows, open a number of novel opportunities for quantum simulation. It makes a convincing case for experimental efforts to implement the proposed setup, making follow-up experimental work viable, and also points to a number of interesting future theoretical avenues to explore.

Thus, I can recommend the manuscript for publication in Communications Physics.

Questions:

- In Section III "State preparation and Detection", the very last paragraph, lines 592-595, briefly discusses how to prepare the superfluid state. Since the section generically makes the claim that the authors can prepare low-energy states, could the authors expand on how to prepare the chiral superfluid (CSF) and density wave ($DW_{\{1/2\}}$, $DW_{\{1/3\}}$) states as well?

- A minor point concerning the presentation in Fig 4. are the interactions shown in Fig 4 (a) truly powerlaw, i.e. are these straight lines on a log-log plot? On a broader point how correct is it to assign a power-law exponent to interactions that are limited to distances of $N-1$ for N internal states? I personally feel that the realisations of tunable longer ranged interactions is significant enough, without the need for a claim/direct comparison to powerlaw interactions, which (at least) from the current presentation may strictly speaking not be present.

I may also be misunderstanding things, but from the expressions in the appendix (D5,D6) I would have expected exponential decay of the coupling constants, rather than power-law.

- In Fig 1 the caption reads "... $j = 0$ site (with dressed state index $m = 0$ and unit cell $\ell = 0$)". I believe this should be dressed state index " $n=0$ "

Version 1:

Reviewer comments:

Reviewer #1

(Remarks to the Author)

The authors have answered my questions and made changes in the manuscript correspondingly. The numerical difficulties mentioned in the reply are understandable. It is good to know that the band gap is sufficiently large such that it is appropriate to discuss long-range interactions without involving higher band physics. I recommend publication of this manuscript in COMMSPHYS.

Reviewer #2

(Remarks to the Author)

The authors have fully addressed the comments and concerns raised in the first round of reviews in their revised manuscript, and I can recommend publication of the revised manuscript.

Specifically:

- Reviewer Point P 1.1 concerning the role of interactions of the CSF: The authors revised the manuscript to note that the interactions are required to determine a unique symmetry broken groundstate

- Reviewer Point P 1.2 concerning order parameters versus order parameter correlation functions: The authors revised the manuscript to explain the role of correlation functions in the context of spontaneous symmetry broken transitions

- Reviewer Point P 1.3 concerning the bandgap: The authors note that their system retains a large bandgap even in parameter regimes where the long-range couplings are strong

- Reviewer Point P 1.4 concerning additional reference: The authors have added the missed references

- Reviewer Point P 2.1 concerning state preparation schemes: The authors have added additional detail on preparing the other ground states in the revised manuscript
- Reviewer Point P 2.2 concerning validity of powerlaw phrasing: the authors have clarified their intent and reasoning for their choice, which I find reasonable
- Reviewer Point P 2.3 concerning typos in figure caption: typo has been corrected

Dear Editor,

We appreciate the thoughtful and constructive input from both reviewers. Overall, we note that Reviewer 2 was quite positive about this work, while Reviewer 1 found the work to be scientifically sound but had a few concerns. Our point-by-point response to each reviewer follows below, along with a summary of changes. We hope the amended manuscript is now ready for publication in Communications Physics.

From the authors,
D. Burba, G. Juzeliūnas, I. B. Spielman, L. Barbiero

Reviewer 1: Specific Comments and Critiques

Burba and coauthors studied long-range tunnelings and long-range interactions in a subwavelength lattice. They found an interesting feature that the signs of long-range tunnelings could be tuned, resulting in geometrical frustration. Since it is typically difficult to engineer geometrical frustration in ordinary optical lattices, the authors' proposal may provide experimentalists with a new useful approach to access geometrical frustration using a relatively simple setup, cyclic Raman couplings. The author further studied many-body phases including a chiral superfluid and some density waves. Whereas I think that the authors' results are interesting and sound, I believe that the authors should address a few issues before publication of this manuscript in Communications Physics.

Reply: We thank the reviewer for her/his report and for the positive assessment of our manuscript. We thank the reviewer also for raising very constructive points that here we accurately address.

Reviewer Point P 1.1 — It seemed that the authors did not provide an explanation for the origin of the chiral superfluid. Does it arise purely from the geometrical frustration or interactions are also necessary? The numerical result of the momentum distribution of this phase shown in Fig. 8 is very interesting since the central peak at zero momentum vanishes while two side peaks arise at finite momenta. As one could imagine that this will happen if the single-particle spectrum has two minima at finite momenta, what happens here? The authors should clarify what is the single-particle spectrum and whether interactions also play a role.

Reply: The reviewer is completely right when claiming that the two peaks at incommensurate k in the momentum distribution signal two minima in the dispersion relation. Notably, this feature is completely general and it applies to both interacting and non-interacting systems. Nevertheless, while at the single-particle level the dispersion relation and the relative Lifschitz transition, i.e. the emergence of two minima, can be exactly calculated, this represents a highly challenging task in many-body quantum systems as exact solutions are usually not available. Specifically, this would require access to a high number of excited states while in matrix product state (MPS) formalism only the very low energy states can be accurately derived. Apart from this technical point, the interaction present in the chiral superfluid (CSF) phase plays the fundamental role of being indeed responsible for the appearance of superfluidity and this being in analogy with normal superfluid (SF) phases. More in detail, in CSF the interaction favors the predominant population of one of the minima and, as a consequence, the system enters a chiral superfluid phase with a non-zero local boson current characterized by a finite chirality $\langle \hat{\kappa}_j \rangle$. In order to clarify this point we added the following text to Subsection 3A "Measurement opportunities":

"The two peaks at incommensurate k in the momentum distribution signal two minima in the dispersion relation. The interaction favors the predominant population of one of the minima and, as a consequence, the system enters a chiral superfluid (CSF) phase with a non-zero local boson current characterized by a finite chirality $\langle \hat{\kappa}_j \rangle$."

Reviewer Point P 1.2 — Chiral often means a finite current, $\langle \hat{\kappa}_j \rangle$, such as the chiral edge state of a topological phase. Whereas the correlation function $\kappa_j^2(\Delta j)$ the authors computed is an important quantity, how about $\langle \hat{\kappa}_j \rangle$ itself?

Reply: Again the reviewer is totally right by claiming that chiral means the presence of a finite current $\langle \hat{\kappa}_j \rangle$, see also our previous reply. In this regard, the long-range of the correlation function $\langle \kappa_j^2 \rangle$ provides the same information, i.e. the chiral order of the systems. As known, phases with a broken discrete symmetry, a Z_2 in this case, can be derived either by calculating the local order parameter capturing the symmetry breaking, here $\langle \hat{\kappa}_j \rangle$, or by proving the long-range order of its relative correlation function $\langle \hat{\kappa}_j \hat{\kappa}_{j+\Delta j} \rangle$. The reason to use this latter rather than the former, relies on the fact that correlation functions are not sensitive to degenerate ground states. Specifically, here the ground state (GS) of the CSF is two fold degenerate signaling that currents can go from right to left or vice versa with the same amplitude. As our MPS simulations correctly find as GS an equally weighted superposition of the two states with equal energy, we find $\langle \hat{\kappa}_j \rangle = 0$ while $\langle \hat{\kappa}_j \hat{\kappa}_{j+\Delta j} \rangle$ is still long-range ordered. A possible strategy, to use the local order parameter instead of the correlator to detect the CSF would imply breaking "by hand" the GS degeneracy by adding a small term $\propto \hat{\kappa}_j$ in the Hamiltonian. Motivated by the fact that this aforementioned and our approach must provide exactly the same result, we preferred to not alter the Hamiltonian and rather focusing on the decay of the correlation function. In order to clarify this point, we added a footnote to Section 2 "Many-body phase diagram", after "The long-range order of $\kappa_j^2(\Delta j)$ indicates correlations between currents on links a distance Δj apart, and is associated with spontaneous breaking of TR symmetry...", which says: "The same conclusions can be derived by calculating directly the order parameter $\hat{\kappa}_j$. Importantly, this strategy requires the addition of a weak term $\hat{\kappa}_j$ in Eq. (14) which allows breaking the ground state degeneracy associated to the currents going from left to right and vice versa with the same amplitude."

Reviewer Point P 1.3 — In ordinary optical lattices, lowering the lattice depth could also increase the amplitudes of long-range interactions due to the increase of the width of the Wannier wavefunctions. But the price to pay is the smaller band gap. As such, when long-range interactions become significant, the band gap may be small enough that p band and even higher bands cannot be ignored anymore. This is the reason that in typical optical lattices, a single-band Hubbard model includes only onsite interactions, not long-range interactions. What happens here to the band gap once the long-range interactions become large enough to induce density waves? If the interactions are comparable to the band gap, one should concern fate of results based on only the lowest band.

Reply: The reviewer is correct, however band gap is significantly larger than interaction strength for our parameter regime, e.g., for $\Omega/E_R = 3.5$, ratio between band gap and interaction strength is $\Delta\epsilon/U \approx 24$. As reviewer rightly mentioned, if band gap wasn't large enough, one would need to consider either higher bands or projected model with renormalized tunnelings and interactions. As lattice depth 2Ω is decreased, band gap $\Delta\epsilon$ also decreases, but, as mentioned previously, band gap is always significantly larger than tunneling and interaction strengths. We also stress that our subwavelength

lattice allows us to reach regime where long-range interactions are non-negligible and band gap is still very large, while for regular optical lattice this would not be the case. Long-range interactions arise in our scheme due to Wannier functions, corresponding to different dressed states, being shifted closer to one another, not due to Schrieffer-Wolff corrections from higher bands, see [74] for examples. To clarify this point, we added the following sentence to Section 2 "Many-body phase diagram": "For smaller Raman couplings, one would need to calculate renormalized Hamiltonian matrix elements, which arise due to higher bands [74]."

Reviewer Point P 1.4 — The authors missed some references about subwavelength lattices, such as PRL 122, 065303 (2019), PRL 123, 260405 (2019), PRL 126, 193001 (2021) PRR 3, 023058 (2021), PRL 128, 173202 (2022).

Reply: We thank the reviewer for pointing out these relevant references. They have been added in Introduction: "We consider the many-body physics of a recently realized class of subwavelength 1D optical lattices...".

Reviewer 2: Specific Comments and Critiques

The manuscript "Many-body phases from effective geometrical frustration and long-range interactions in a subwavelength lattice" demonstrates that Raman-induced sub-wavelength lattices allow to realise Hubbard models with frustrated long-range tunnel couplings, and long-range density-density interactions. It then shows via numerical simulations that this results in a range of many-body phases, namely density wave, superfluids and chiral superfluids. The manuscript finally provides a protocol to measure observables characterising these phases, which are defined in a dressed state basis, thus, making them experimentally observable.

It therefore makes a number of significant contributions and advances. (i) it realises frustrated long-range tunable tunnel couplings/geometric frustration, typically not (easily) accessible in optical lattices, (ii) it realises long-range repulsive interactions in a system with nominally only contact interactions, (iii) it establishes a range of many-body phases, and how to observe them experimentally.

The results are well presented and fully supported by the theoretical model and equations, the accompanying figures, and the (numerically-exact) simulation methods, and are fully convincing. The results are appropriately discussed in the context of the prior literature. Technical details are appropriately discussed in detail in the supplementary material, allowing reproduction of the results.

The work is timely and of great interest to the wider community, as sub-wavelength lattices are only beginning to be explored, and as this work shows, open a number of novel opportunities for quantum simulation. It makes a convincing case for experimental efforts to implement the proposed setup, making follow-up experimental work viable, and also points to a number of interesting future theoretical avenues to explore.

Thus, I can recommend the manuscript for publication in Communications Physics.

Reply: We thank the reviewer for her/his report and for the positive assessment of our manuscript. We thank the reviewer also for raising very constructive points that here we accurately address.

Reviewer Point P 2.1 — In Section III "State preparation and Detection", the very last paragraph, lines 592-595, briefly discusses how to prepare the superfluid state. Since the section generally makes the claim that the authors can prepare low-energy states, could the authors expand on how to prepare the chiral superfluid (CSF) and density wave ($DW_{1/2}$, $DW_{1/3}$) states as well?

Reply: Once the superfluid state is prepared, the chiral superfluid (CSF) and density wave ($DW_{1/2}$, $DW_{1/3}$) states can be accessed through adiabatic ramps that modify the system parameters to the required regimes. This method mirrors the well-established techniques used for superfluid-to-Mott insulator transitions in one, two, and three dimensions. To clarify this point, the following sentence was added to Subsection 3B "Staggered readout": "Finally, once the superfluid state is prepared, the chiral superfluid (CSF) and density wave ($DW_{1/2}$, $DW_{1/3}$) states can be accessed through adiabatic ramps that modify the system parameters to the required regimes."

Specifically, these ramps can be executed over a timescale of approximately 50 ms, which is significantly shorter than the $1/e$ lifetime due to spontaneous emission mentioned in our work: "In the present work, spontaneous emission from the Raman lasers is the only intrinsic heating process...".

While finite ramp rates across phase transitions inevitably create excitations, our approach benefits from native energy scales comparable to U , which ensures excitation levels akin to those observed in traditional superfluid-Mott insulator (SF-MI) transitions. A detailed characterization of the excitations generated during these ramps remains a compelling avenue for future study.

Reviewer Point P 2.2 — A minor point concerning the presentation in Fig 4. are the interactions shown in Fig 4 (a) truly powerlaw, i.e. are these straight lines on a log-log plot? On a broader point how correct is it to assign a power-law exponent to interactions that are limited to distances of $N-1$ for N internal states? I personally feel that the realisations of tunable longer ranged interactions is significant enough, without the need for a claim/direct comparison to powerlaw interactions, which (at least) from the current presentation may strictly speaking not be present. I may also be misunderstanding things, but from the expressions in the appendix (D5,D6) I would have expected exponential decay of the coupling constants, rather than power-law.

Reply: Interactions do not truly follow power law - power law is an approximation. This was mentioned in Subsection 1D "Comparison to other long-range interactions": "Our scheme is not subject to this limitation and α is not fixed a priori.". The reviewer is correct that for sufficiently large distances, one indeed gets exponential decay of interaction strengths, which was explained in Appendix D. Power law fit is useful for intuition and to compare to other works since power law tunnelings and interactions have been widely studied. For ground state, agreement with power law only matters for a few shorter range Hamiltonian matrix elements (tunnelings and interactions) and thus our fit is justified.

Reviewer Point P 2.3 — In Fig 1 the caption reads "... $j = 0$ site (with dressed state index $m = 0$ and unit cell $l = 0$)". I believe this should be dressed state index " $n=0$ ".

Reply: Indeed there was a typo and we fixed it to be the following: "... (with dressed state $n = 0$)...".

Summary of Changes

- In response to Point 1.1 from Reviewer 1, we added the following text to Subsection 3A "Measurement opportunities": "The two peaks at incommensurate k in the momentum distribution signal two minima in the dispersion relation. The interaction favors the predominant population of one of the minima and, as a consequence, the system enters a chiral superfluid (CSF) phase with a non-zero local boson current characterized by a finite chirality $\langle \hat{\kappa}_j \rangle$."
- In response to Point 1.2 from Reviewer 1, we added a footnote to Section 2 "Many-body phase diagram", after "The long-range order of $\kappa_j^2(\Delta j)$ indicates correlations between currents on links a distance Δj apart, and is associated with spontaneous breaking of TR symmetry...", which says: "The same conclusions can be derived by calculating directly the order parameter $\hat{\kappa}_j$. Importantly, this strategy requires the addition of a weak term $\hat{\kappa}_j$ in Eq. (14) which allows breaking the ground state degeneracy associated to the currents going from left to right and vice versa with the same amplitude."
- In response to Point 1.3 from Reviewer 1, we added the following sentence to Section 2 "Many-body phase diagram": "For smaller Raman couplings, one would need to calculate renormalized Hamiltonian matrix elements, which arise due to higher bands [74]."
- In response to Point 1.4 from Reviewer 1, we have added additional references in the Introduction: "We consider the many-body physics of a recently realized class of subwavelength 1D optical lattices..."
- In response to Point 2.1 from Reviewer 2, the following sentence was added to Subsection 3B "Staggered readout": "Finally, once the superfluid state is prepared, the chiral superfluid (CSF) and density wave ($DW_{1/2}$, $DW_{1/3}$) states can be accessed through adiabatic ramps that modify the system parameters to the required regimes."
- In response to Point 2.3 from Reviewer 2, we have fixed a typo in figure caption: "... (with dressed state $n = 0$)..."